# Prediction of Maxillary Bone Invasion in Hard Palate/Upper Alveolus Cancer: A Multi-Center Retrospective Study

**DOI:** 10.3390/cancers15194699

**Published:** 2023-09-24

**Authors:** Nayeon Choi, Jeon Yeob Jang, Min-Ji Kim, Sung Seok Ryu, Young Ho Jung, Han-Sin Jeong

**Affiliations:** 1Department of Otorhinolaryngology-Head and Neck Surgery, Sungkyunkwan University School of Medicine, Samsung Medical Center, Seoul 06351, Republic of Korea; chlskduschoi@naver.com; 2Department of Otolaryngology, Ajou University Hospital, Ajou University School of Medicine, Suwon 16499, Republic of Korea; manup1377@gmail.com; 3Biomedical Statistics Center, Research Institute for Future Medicine, Samsung Medical Center, Seoul 06351, Republic of Korea; rabbit93.kim@samsung.com; 4Department of Otolaryngology, Ulsan University School of Medicine, Asan Medical Center, Seoul 05505, Republic of Korea; ryu1933@naver.com

**Keywords:** hard palate, alveolar process, neoplasm, bone, diagnostic imaging

## Abstract

**Simple Summary:**

Pathological bone invasion is an independent, poor prognostic factor in oral cancer, and accurate prediction of bone invasion is critical to the prognosis estimation and treatment decision. Many previous studies on bone invasion of oral cancer have focused on mandibular invasion, but there have been relatively few reports about the maxillary bone invasion (MBI) of hard palate/upper alveolus (HP/UA) cancer. Therefore, we have attempted to design a prediction model for MBI using several radiological and clinical variables of HP/UA cancer. We found that computerized tomography (CT) alone predicted MBI, with a discrimination ability of 77.9%. Meanwhile, the discrimination performance was increased up to 91.1% in a prediction model including CT findings, tumor dimensions, clinical factors (male sex, nodal metastasis), and maximal standardized uptake value of positron emission tomography/CT. In addition, the scoring system using these variables clearly distinguished low-, intermediate-, and high-risk groups for MBI in HP/UA cancer.

**Abstract:**

Background: maxillary bone invasion (MBI) is not uncommon in hard palate or upper alveolus (HP/UA) cancer; however, there have been relatively few reports about the MBI of HP/UA cancer. Patients and Methods: this was a multi-center retrospective study, enrolling 144 cases of HP/UA cancer. MBI was defined by surgical pathology or radiology follow-up. The multiple prediction models for MBI were developed in total cases and in cases having primary bone resection, using clinical and radiological variables. Results: computerized tomography (CT) alone predicted MBI, with an area under receiver operating curve (AUC) of 0.779 (95% confidence interval (CI) = 0.712–0.847). The AUC was increased in a model that combined tumor dimensions and clinical factors (male sex and nodal metastasis) (0.854 (95%CI = 0.790–0.918)). In patients who underwent ^18^fluorodeoxyglucose positron emission tomography/CT (PET/CT), the discrimination performance of a model including the maximal standardized uptake value (SUVmax) had an AUC of 0.911 (95%CI = 0.847–0.975). The scoring system using CT finding, tumor dimension, and clinical factors, with/without PET/CT SUVmax clearly distinguished low-, intermediate-, and high-risk groups for MBI. Conclusion: using information from CT, tumor dimension, clinical factors, and the SUVmax value, the MBI of HP/UA cancer can be predicted with a relatively high discrimination performance.

## 1. Introduction

Pathological maxillary bone invasion (MBI) is an independent, poor prognostic factor in oral cavity (OC) squamous cell carcinoma [1,2,3,4,5,6,7]. MBI of hard palate and upper alveolus (HP/UA) cancer can result in an increase in the extent of surgery, frequently requiring reconstruction, and leading to functional and cosmetic complications after definitive treatment [8]. A palatal obturator (prosthesis), instead of reconstruction, is one of the attractive management options for oronasal fistula after definitive treatments for HP/UA cancer. However, unfitting, gravitational displacement, deformation, irritation, or hygiene problems may occur in some patients [9]. Many previous studies on bone invasion of OC cancer have focused on mandibular invasion [10,11,12,13,14,15], but there have been relatively few reports about the MBI of HP/UA cancer [16,17].

Generally, HP/UA cancers have shown better oncologic outcomes than OC cancers of other locations [7,18,19,20,21,22]. HP/UA cancer has distinct characteristics in comparison with oral tongue and buccal cancer because of its proximity to the maxillary bone of the HP. The maxilla has a lower bone density than the mandible and is considered to be porous [23]. The tumors in this region tend to invade adjacent tissues as they grow [5]. The maxillary bone of the HP is much thinner than the mandible, and it is difficult to identify clearly using preoperative imaging modalities whether tumors invade through the maxillary bone [15,17].

Contrast-enhanced computed tomography (CT) has been recognized as the gold standard for the evaluation of bone invasion in OC cancer [12,13,17], and several other imaging modalities including positron emission tomography (PET)/CT, magnetic resonance imaging (MRI), and single-photon emission CT also have been reported to be effective for preoperative determination of bone invasion [10,11]. However, none of these imaging modalities have proven to have sufficient diagnostic accuracy in HP/UA cancer. Therefore, in this study, we aimed to design statistical models that can predict the MBI of HP/UA cancer by combining several preoperative clinical and radiological factors.

## 2. Materials and Methods

### 2.1. Study Populations

This research was a multi-center retrospective study, which was approved by the authors’ Institutional Review Board (Samsung Medical Center, Asan Medical Center, and Ajou University Hospital, Republic of Korea). We enrolled patients with biopsy-proven HP/UA cancer who underwent curative surgical resection between 2005 and 2020, and excluded patients with other OC subsites such as the buccal and oral tongue. The pathological diagnosis consisted of squamous cell carcinoma, salivary gland carcinoma, and melanoma. Tumor pathology and histologic grade were classified according to the 5th edition of the World Health Organization histological classification [24], and tumor staging was defined based on the American Joint Committee on Cancer (8th edition) Tumor–Node–Metastasis Staging Manual. The initial number of enrolled cases from the three hospitals was 177, and 33 were excluded due to the lack of adequate medical information. The remaining 144 cases were analyzed in this study (84 from Samsung Medical Center, 46 from the Asan Medical Center, and 14 from Ajou University Hospital). Study characteristics were not different among the three medical centers (Appendix A).

### 2.2. Preoperative Examinations and Analyzed Clinical Factors

All patients received biopsies of their lesions for pathological diagnosis. Head and neck contrast-enhanced CT and PET/CT were performed preoperatively. If there was a suspicious cervical lymph node metastasis, fine needle aspiration or core needle biopsy was performed. MRI was also performed, particularly in cases with extensive tumor infiltration or nerve-related symptoms including pain, numbness, and paralysis. After the preoperative evaluation, all patients underwent wide resection of the primary tumor (with underlying bone resection = 71, without bone resection = 73), and neck dissection according to imaging and histological diagnosis, if indicated.

Clinical bone invasion was estimated by preoperative imaging and physical examination. The baseline characteristics including age, sex, and tumor-related physical factors including tumor long axis, short axis, and thickness were analyzed. The diameters of the tumor long axis and short axis, and the thickness of the surgical specimens were measured, and the tumor area was calculated (π × tumor long axis/2 × tumor short axis/2). The maximal standardized uptake value (SUVmax) of the primary tumors on PET/CT was also recorded (Figure 1).

### 2.3. CT and PET/CT Scans Protocols and Evaluation of Clinical Bone Invasion

Contrast-enhanced CT scans (LightSpeed Ultra or Ultra 16, GE, Milwaukee, WI, USA) of the head and neck were performed with the following parameters. The sliced section width was 3.75 mm with 160 mAs, 120 KeV, and a table feed rate of 8.75 mm per rotation. The iodinated contrast agent (Ultravist 300, Schering, Berlin, Germany) was injected intravenously at 3 mL/s for contrast enhancement, and the scan delay time was 30 sec. Radiological CT bone invasion was defined as when a contrast-enhanced primary tumor was identified within the cortical bone, and the cortical bone was partially eroded or totally destroyed.

^18^Fluorodeoxyglucose (FDG)-PET/CT scans (GE Discovery LS scanner, GE, Milwaukee, WI, USA) were combined with PET and CT scans without contrast enhancement. All patients fasted for 6 h before the exam. Whole-body CT scans were carried out using a continuous spiral technique and an 8-slice helical CT with a gantry rotation speed of 0.8 s. CT scans were performed with the following settings: 40–80 mAs, 140 KeV, a section width of 5 mm, and a table feed of 5 mm/rotation. After intravenous injection of 370 MBq FDG, an emission scan was taken from thigh to head at 5 min/frame for 45 min. CT and FDG-PET scan data were accurately conjugated using commercial software (eNTEGRA Workstation R 2.0, GE Medical Systems, Milwaukee, WI, USA). The SUVs were acquired using attenuation-corrected images, adjusting the amount of FDG injected, patient body weight, and cross-calibration factors between FDG-PET and the dose calibrator.

The CT and FDG-PET/CT images were reviewed by two radiologists with more than 10 years of clinical practice in head and neck radiology, and two experienced nuclear medicine physicians with more than 10 years of PET/CT interpretation. Each observer independently reviewed the images in a randomized fashion, and the observers reached consensus by joint interpretation.

### 2.4. Pathology Evaluation

The hematoxylin and eosin-stained surgical specimen slides were reviewed, and a determination of the histological pattern of MBI was made for each specimen by two experienced pathologists of each medical center, who had more than 10 years of clinical experience in head and neck pathology. Any discrepancy was solved by joint discussion by the two pathologists. Pathological MBI was determined when the invasion of malignant cells into cortical bone or cancellous bone of the medullary cavity was identified (Figure 1). Additional immuno-staining was performed for pathological subtyping, if indicated.

### 2.5. Development of Prediction Models for MBI

We defined pathological MBI (*n* = 46) as the gold standard for true positivity, in cases with primary bone (HP/UA) resection (*n* = 71). Absence of MBI in bone-resected cases was set as true negativity for MBI (*n* = 25). In addition, for cases without bone resection (*n* = 73), the absence of MBI was confirmed by the clinical and radiological follow-up of more than 2 years (*n* = 53). Clinical and radiologic bone invasion during follow-up was detected in 24 patients. The final 70 patients were defined as MBI-positive (46 with bone resection and 24 without bone resection).

The multiple prediction models for MBI were developed in total cases (*n* = 144) and confirmed in cases having primary bone resection (*n* = 71). Firstly, we selected significant variables using univariable logistic regression analysis and determined cut-off values of the continuous variables (tumor long axis, tumor area, PET SUVmax) using receiver operating characteristics (ROC) curve analysis. After selection of clinical and radiological variables based on a *p*-value < 0.05 of the univariable analysis, we constructed prediction models with a combination of selected variables using logistic regression analysis. ROC curves of each prediction model were plotted, and the AUC (area under curve) for each curve was calculated to evaluate the discrimination performance of the prediction model and compared with DeLong’s test. The final models for MBI were validated by use of five-times-repeated five-fold cross-validation [25]. The performance of each prediction model was also demonstrated with a Brier score (squared difference between actual outcome and prediction), where smaller values indicate better overall performance [26,27].

### 2.6. Design of a Predictive Scoring System for MBI

After developing prediction models using logistic regression analysis, we established a score-based predictive system for MBI to enhance the clinical applicability. We assigned scores to each predictor variable by dividing their beta coefficients by the absolute value of the smallest coefficient in the final model. Then, we rounded up the results to the nearest integer, generating a simple integer-based point score for each predictor [28]. Next, we calculated the total score for each patient by adding up the scores of all predictor variables, and used a ROC curve to assess the predictive value of our scoring system for MBI. Additionally, we internally validated the rule by employing the bootstrap method on the original dataset, which involved sampling with replacement for 1000 iterations.

### 2.7. Statistical Analyses

We compared clinical characteristics between groups without and with MBI. The categorical variables were compared using Fisher’s exact test and the continuous variables were compared using the Mann–Whitney U test. All tests were two-sided, and a *p*-value of less than 0.05 was considered to be statistically significance. Statistical analyses were performed using SPSS software (IBM SPSS Statistics ver. 21, Chicago, IL, USA) and STATA 12 (STATA, College Station, TX, USA), and SAS version 9.4 (SAS Institute, Cary, NC, USA).

## 3. Results

### 3.1. Comparison of Clinical Characteristics between the No MBI and MBI Groups

Comparisons of the no MBI and MBI groups are presented in Table 1. The age of the no MBI (56.2 ± 15.3 years) and MBI (57.2 ± 14.7 years) groups was not significantly different (*p* = 0.573), but male predominance was shown in the MBI group (*p* = 0.030) compared with the no MBI group. Tumor location and histological diagnosis were not different between the two groups (*p* = 0.152 and *p* = 0.283, respectively).

Using CT, MBI was estimated in 20 (27.0%) patients of the no MBI group and 58 (82.9%) patients of the MBI groups, which were significantly different (*p* < 0.001). The SUVmax on PET-CT was significantly higher (*p* = 0.003) in the MBI group (9.5 ± 4.6) than in the no MBI group (6.6 ± 4.9). Tumor diameter, thickness, and area were significantly higher (*p* < 0.001) in the MBI group. Clinical lymph node staging was also higher in the MBI group (*p* = 0.042).

### 3.2. AUC and Univariable Logistic Regression Analysis of Clinical Variables for MBI

We identified significant variables for the prediction of MBI (Table 2). In the univariate logistic regression analysis, sex (male), CT bone invasion, PET SUVmax, the diameters of the long and short axes, thickness, area of the tumor, and LNM were shown to be significant risk factors. In contrast, age, tumor location, and histological type were not significant risk factors for MBI. In addition, the AUCs of significant variables in univariable analysis were higher than non-significant variables.

### 3.3. MBI Prediction Models by Multivariable Logistic Regression Analysis

We used combinations of significant variables on univariable and AUC analysis to develop several prediction models for MBI of HP/UA cancer (Table 3 and Figure 2). The subsequent models were designed by excluding cases with missing data of the selected variables.

Model 1 was developed with an image variable only (CT bone invasion), and had an AUC of 0.779 (95% confidence interval, 95%CI = 0.712–0.847). It had the lowest performance (Brier score = 0.168). Model 2 included four variables (CT, tumor long axis diameter, sex, and presence of clinical nodal metastasis), and it showed better predictive performance (AUC = 0.853, 95%CI = 0.790–0.917, Brier score = 0.152) than model 1. Instead of tumor long axis diameter, tumor area was included as a tumor dimension variable in model 3 (CT, tumor area, sex, and presence of clinical nodal metastasis). Model 3 had an AUC of 0.854 (95%CI = 0.790–0.918), and a Brier score of 0.152, and the results were similar to those of model 2.

Lastly, we included the PET SUVmax variable in model 4 (CT, tumor long axis diameter, sex, and presence of clinical nodal metastasis, and PET SUVmax). Model 4 had the best predictive performance, with an AUC of 0.911 (95%CI = 0.847–0.975), and a Brier score of 0.110. The differences in the AUCs in each model were also significant, as indicated by DeLong’s test.

### 3.4. Sensitivity Analysis in Cases with Primary Bone (HP/UA) Resection

Prediction models 1–4 were based on the findings of total patients (*n* = 144), and true positivity and negativity for MBI were determined by surgical pathology or radiology follow-up. Next, we tested whether our prediction models were also valid in cases with primary bone (HP/UA) resection (*n* = 71), where surgical pathology was the gold standard for MBI (Appendix A).

Model 5 included four variables (CT, tumor area, sex, and lymph node metastasis), and it showed an AUC of 0.8045 (95%CI = 0.6879–0.9210). DeLong’s test showed a significant difference between model 1 and 5 (*p* = 0.0310). Model 6 was established for patients with PET-CT who underwent bone resection during initial surgery (*n* = 44). Model 6 included five variables (CT, tumor area, sex, lymph node metastasis, and PET SUVmax) and it had a better predictive performance, with an AUC of 0.8548 (95%CI = 0.7027–1.0000), than that of model 5. In models 5 and 6, the Brier score was not calculated due to the low AUC.

### 3.5. Validation and Calibration of Predicted Models

The final model for MBI was confirmed by internal validation using five times-repeated five-fold cross-validation. The validated AUCs of each model were 0.839 (95%CI = 0.687–0.983), 0.844 (95%CI = 0.695–0.984), 0.884 (95%CI = 0.758–0.993), 0.7328 (95%CI = 0.482–0.973) and 0.712 (95%CI = 0.345–0.990) for models 2, 3, 4, 5, and 6, respectively (Table 3 and Appendix A). In addition, the Brier score decreased from model 1 to model 4, which meant that model 4 had the best diagnostic performance. Calibration of the prediction models (model 2, 3, and 4) was assessed graphically, in a plot with the observed proportions of PBI and the number of patients, according to deciles of predictions. The correspondence was close between the deciles of predictions and the observed proportions, indicating that our models were well calibrated.

### 3.6. Scoring Systems for MBI Prediction and Assigned Points

Scoring systems were established based on logistic regression analysis, and the score for each variable was determined based on the regression coefficient ratio (Table 4). The scoring systems were made up of two versions depending on the inclusion or exclusion of PET SUVmax. Scores were calculated as follows: 5 points for bone invasion on CT, 2 points for tumor area more than 4.0 cm^2^ and male, 1 point for clinical lymph node metastasis, and 3 points for PET SUVmax more than 6.0. The patients were grouped into low (0–4 points), intermediate (5–7 points), and high risk (8–10 points) by the scoring system without PET SUVmax. Otherwise, the patients were classified into low (0–6 points), intermediate (7–8 points), and high risk (9–13 points) by the scoring system with PET SUVmax.

The incidence of MBI using each classification of the scoring system was described (Figure 3). The scoring system using preoperative CT finding, tumor dimension, and clinical factors (male sex and node metastasis), with/without PET SUVmax clearly distinguished low-, intermediate- and high-risk groups for MBI. By conducting this validation process, we ensured that the scoring system was reliable and effective in predicting maxillary bone invasion. The ROC curve provided valuable insights into the system’s performance, and the bootstrap method confirmed its stability and generalizability.

## 4. Discussion

MBI of the HP/UA bone is an independent risk factor for poor oncological outcomes in OC cancer [7,16,17,18,19,20,21,22]. Moreover, the complete surgical resection and reconstruction are highly dependent on the preoperative evaluation of the disease extent. If there is minimal risk of MBI in the HP/UA cancer, the surgical resection usually includes the mucoperiosteum of the bone, with the underlying bone left intact. In this case, the overlying mucosa is healed secondarily without reconstruction. In contrast, in cases when PBI is suspected, the surgical resection of HP/UA cancer involves soft-tissue or bone reconstruction for the surgical defect. A palatal obturator (prosthesis), instead of reconstruction, is one of the management options for the surgical defect of the HP/UA, but unfitting, displacement, deformation, irritation, or hygiene problems may occur in some patients [9]. Therefore, it is essential to predict the presence or extent of MBI of UP/UA cancer accurately, to achieve the best surgical outcomes. Although there have been numerous reports on the prediction and management of mandible invasion from OC cancer [10,11,12,13,14,15], the studies on MBI of HP/UA cancer are currently insufficient.

It is well known that the CT scan is the best method to evaluate the bone invasion of OC cancer, including HP/UA cancer [26,29,30,31]. However, we found that the discrimination ability for HP/UA bone invasion was not sufficiently high (AUC 77.9%, 95%CI = 71.2–84.7%). In this study, we aimed to design a more accurate (>90%) prediction tool for MBI of HP/UA cancer using a number of preoperative parameters. As a result, we improved the discrimination accuracy with a prediction model using multiple clinical and PET/CT factors in addition to CT (AUC 91.1%, 95%CI = 84.7–97.5%). Our findings seem clinically meaningful, with more than a 10% increase in discrimination accuracy for MBI of HP/UA cancer.

According to a previous study, the diagnostic performance of the CT scan for bone invasion was a sensitivity, specificity, and diagnostic accuracy of 80%, 100%, and 89.4%, respectively [29]. A meta-analysis showed that the CT scan for mandibular invasion had a sensitivity and specificity of 72% and 90%, respectively, and an AUC of 0.90 [30]. However, these studies evaluated mandible invasion of OC cancer with CT scans. The detection of bone invasion with CT showed relatively low diagnostic performance for HP/UA cancer-related MBI compared with mandible invasion. Thickness and bone mineral density were lower in the maxilla than in the mandible, which could be associated with the diagnostic differences in the CT evaluation [23]. Supporting this concept, one study on HP/UA cancer demonstrated a sensitivity, specificity, and accuracy of 80%, 88%, and 86%, respectively, for the CT detection of MBI [17]. In this study, we found that CT alone had 71.2–84.7% discrimination power for MBI of HP/UA cancer.

A comparative study of MRI, CT, and PET/CT for the detection of mandibular invasion showed that MRI had the highest sensitivity of the three [15,31]. The authors concluded that any single imaging modality did not have sufficient diagnostic accuracy, and multiple imaging methods should be considered together for accurate prediction. Therefore, it is necessary to improve the diagnostic accuracy of the CT scan in conjunction with multiple imaging and clinical variables. Likewise, MRI could improve the diagnostic ability of MBI with its higher sensitivity in HP/UA cancer. In this study, MRI was performed only in a subset of our cohort (*n* = 33), and we did not analyze the diagnostic ability of MRI for HP/UA bone invasion. In this study, MRI was performed only in cases where there was extensive disease or suspicious bone invasion in CT images. Consequently, those who underwent a preoperative MRI also underwent primary bone resection, and most of them were diagnosed with pathologic bone invasion. This introduced a significant selection bias, which led us to exclude MRI from our analysis. Therefore, this point should be re-evaluated in future studies.

Along with CT scans, PET/CT has been investigated in several studies for predicting bone invasion [10,13,14,15]. The SUVmax of PET/CT stratified the risk of bone invasion, which was 53.6% for tumors with a SUVmax of 9.5–14.5, and 71.4% with a SUVmax above 14.5 [10]. The cut-off value of the SUVmax = 9.5 showed a high sensitivity of 97.6%, but a low specificity of 31.2% [10]. The diagnostic sensitivity for bone invasion was highest with PET/CT; meanwhile, the specificity was best with the CT scan [13]. This was because of the relatively high false-positive rate with PET/CT (for example, high glucose uptake in inflammation). Furthermore, glucose uptake cannot be significantly increased in patients with small tumor volume and minimal bone invasion [13]. Therefore, a combination of structural and metabolic imaging may be better for the prediction of MBI of HP/UA cancer. In our study, the prediction model including the SUVmax of PET/CT showed the highest AUC of 0.911.

Metabolic tumor imaging with PET/CT has been suggested as a surrogate marker for tumor aggressiveness [32,33]. Aggressive tumor growth reflects various genotypic and phenotypic features of the tumor, and it also results in the enhancement of the glycolytic metabolism of cancer cells exposed to hypoxia [34,35], which increases the glucose uptake of the tumor in PET/CT [36,37]. This might be correlated with tumor aggressiveness and more bone invasion of OC cancer. Similarly, neck metastasis can also reflect the aggressiveness of the primary tumor. In our study, clinical nodal metastasis was significantly more common in HP/UA cancer with MBI (odds ratio 2.417, *p* = 0.018). The prediction model for MBI of HP/UA cancer was improved in terms of discrimination performance by adding information about clinical lymph node metastasis. This finding was in line with a previous study, which showed that bone invasion was significantly associated with cervical lymph node metastasis [38].

The tumor dimension (size) is also a well-known poor prognostic factor in OC cancer [38,39], and a correlation between tumor dimension and MBI could be assumed. In our study, we tested various tumor dimension variables, such as the long diameter of the tumor, tumor surface area, tumor thickness (in pathology), and tumor volume. However, preoperative measurement of tumor thickness (i.e., depth) from clinical examination or radiological images demonstrated problems with inaccuracy, such that it seemed clinically unpractical. In contrast, the long diameter of tumors and tumor area (with a short diameter) can be easily measured in clinical practice. Therefore, we constructed the subsequent models (models 2–4) using the tumor long axis (cut-off value = 1.8 or 2.0 cm) and tumor area (cut-off value = 4.0 cm^2^). In the final model, the discrimination power was further improved by incorporating the tumor dimension variable into the model.

Another thing to note was that the proportion of male patients with HP/UA cancers with MBI was significantly higher than those without MBI (55.7% vs. 37.8%, respectively, *p* = 0.030). In logistic regression analysis, male patients also showed a higher risk of MBI than female patients (odds ratio = 2.067, *p* = 0.033). The reason for this was not clear, but tumors in male patients seemed more aggressive than those in female patients. In addition, the type of histology did not show any significant difference regarding the MBI of HP/UA cancer in our study. However, the number of pathological subtypes (for example, minor salivary gland cancer) was not sufficient in our study. These findings also should be confirmed in a future large-scale study.

There were several limitations in this study. In cases without primary bone resection (HP/UA), we defined the true negative cases simply by radiological and clinical follow-up, which could cause an underestimation of the true MBI, particularly in patients with adjuvant therapy. To overcome this limitation partially, we conducted a secondary analysis of the cases with primary bone resection (models 5 and 6 in Supplementary Table 2), and confirmed our results in total cases (models 3 and 4 in Table 3). In addition, we did not analyze the diagnostic ability of MRI for HP/UA MBI, because of the limited number of cases. Therefore, further prospective studies are required to support our conclusion. Even with these study limitations, we analyzed a relatively large number of subjects (from three centers) with homogeneous tumor locations (HP/UA) in our study. In addition, we developed a more accurate prediction model for MBI in HP/UA cancer, using multiple imaging and clinical parameters, as well as a CT scan.

## 5. Conclusions

In this study, we designed a prediction model for MBI of HP/UA cancer, using information from CT, tumor dimension, clinical factors, and the SUVmax value, and demonstrated that MBI of HP/UA cancer can be predicted with a relatively high discrimination performance. We expect that our results could improve the surgical planning for HP/UA cancer and provide more accurate information to clinicians and patients.

## Figures and Tables

**Figure 1 cancers-15-04699-f001:**
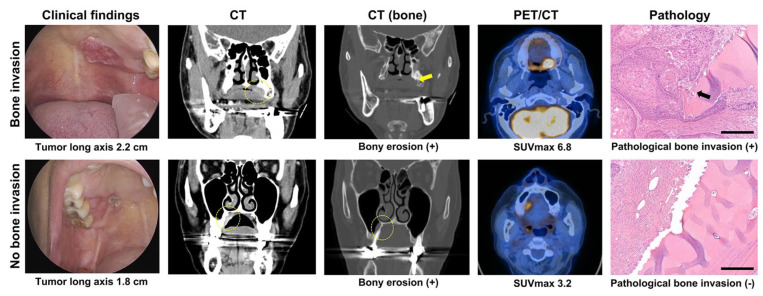
Representative images of bone invasion of HP/UA malignancy. (**Upper panel**) Sixty-five-year-old male with squamous cell carcinoma of HP showed primary tumor long axis of 2.2 cm with PET-SUVmax of 6.8 and suspicious bone erosion on CT (yellow arrow). Cortical bone invasion of maxillary bone (black arrow) was identified by pathology (scale bar = 300 μm). (**Lower panel**) Sixty-six-year-old female with squamous cell carcinoma of UA demonstrated primary tumor long axis of 1.8 cm with PET-SUVmax of 3.2 and suspicious bone invasion on CT. No pathological bone invasion was confirmed by pathology.

**Figure 2 cancers-15-04699-f002:**
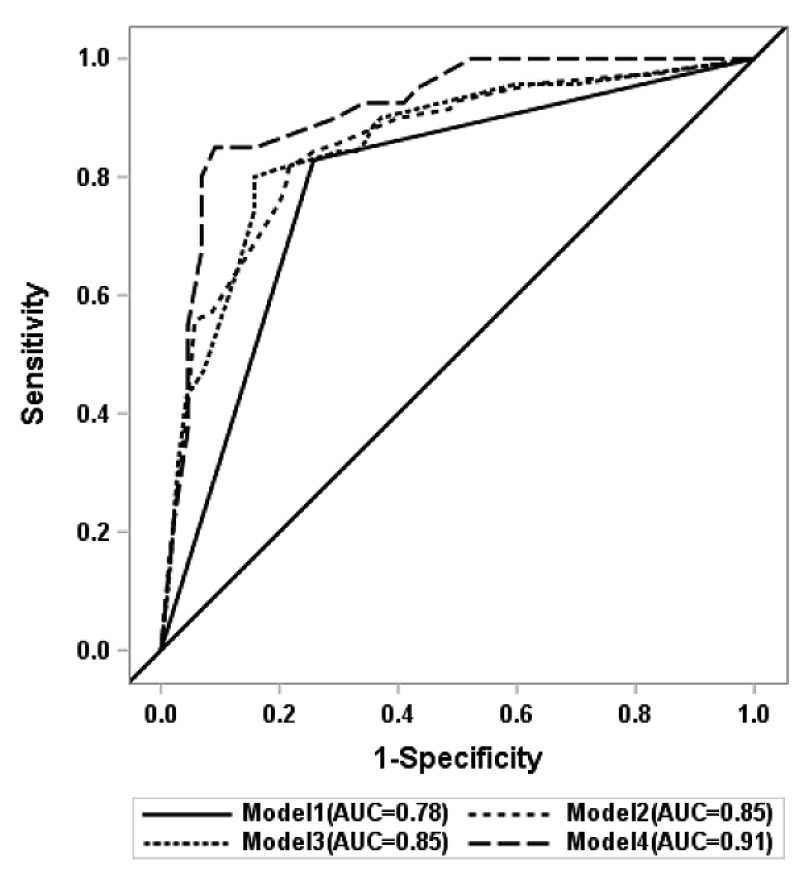
ROC curves of the prediction models. Model 1: CT bone invasion. Model 2: CT bone invasion + tumor long axis + male sex + clinical lymph node metastasis. Model 3: CT bone invasion + tumor area + male sex + clinical lymph node metastasis. Model 4: CT bone invasion + tumor long axis + male sex + clinical lymph node metastasis + PET SUVmax.

**Figure 3 cancers-15-04699-f003:**
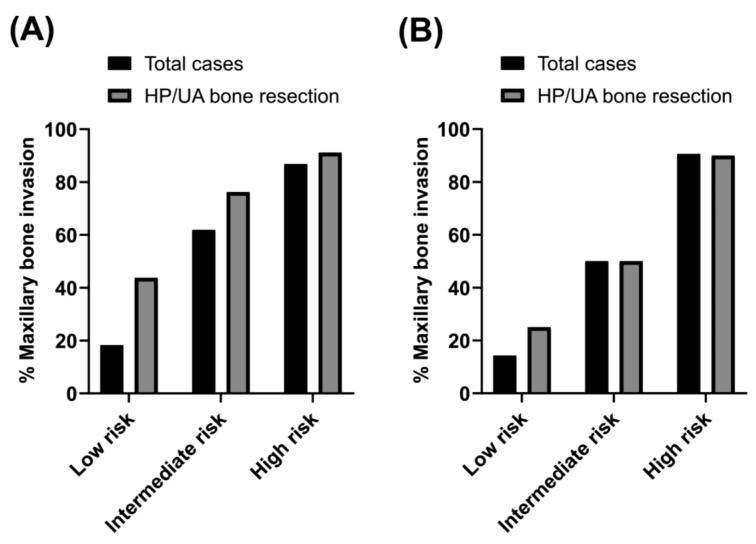
Scoring systems for the prediction of pathologic bone invasion. The black bars represent total patients, and the gray bars represent patients who underwent primary bone resection during the initial surgery. (**A**) The scoring system constructed by CT, tumor area, sex (male), and clinical lymph node metastasis demonstrated the proportion of pathological MBI and risk stratification. The risk classifications were categorized as follows: low risk (0–4), intermediate risk (5–7), and high risk (8–10). (**B**) The scoring system constructed by CT, tumor area, sex (male), clinical lymph node metastasis, and PET SUVmax showed the proportion of pathological MBI and risk stratification. The risk classifications were classified into low (0–6), intermediate (7–8) and high risk (9–13) for MBI.

**Table 1 cancers-15-04699-t001:** Comparison of clinical characteristics between the patients without and with maxillary bone invasion (*n* = 144).

	No Bone Invasion	Bone Invasion	*p-*Value
Patient No. (%)	74 (51.4)	70 (48.6)	
Age (years, mean ± SD)	56.2 ± 15.3	57.3 ± 14.7	0.573
Sex (male: female) (No., %)	28:46 (37.8:62.2)	39:31 (55.7:44.3)	0.030
Tumor location (No., %)			
Hard palate	64 (86.5)	54 (77.1)	0.152
Upper alveolus	10 (13.5)	16 (22.9)	
Histology (No., %)			
Squamous cell carcinoma	27 (36.5)	33 (47.1)	0.283
Minor salivary gland carcinoma	37 (50.0)	32 (45.7)	
Mucoepidermoid carcinoma	19 (25.7)	12 (17.1)	
Adenoid cystic carcinoma	5 (6.8)	14 (20.0)	
Carcinoma ex pleomorphic adenoma	4 (5.4)	0 (0)	
Adenocarcinoma	3 (4.1)	5 (7.1)	
Epithelial-myoepithelial carcinoma	3 (4.1)	0 (0)	
Acinic cell carcinoma	2 (2.7)	1 (1.4)	
Salivary duct carcinoma	1 (1.4)	0 (0)	
Melanoma	10 (13.5)	5 (7.1)	
Bone invasion on CT (No., %)	20 (27.0)	58 (82.9)	<0.001
PET SUVmax (mean ± SD)	6.6 ± 4.9	9.5 ± 4.6	0.003
Tumor long axis (cm, mean ± SD)	2.1 ± 1.5	3.3 ± 1.3	<0.001
Tumor short axis (cm, mean ± SD)	1.5 ± 1.1	2.5 ± 1.2	<0.001
Tumor thickness (cm, mean ± SD)	1.1 ± 0.9	1.8 ± 1.0	<0.001
Tumor area (cm^2^, mean ± SD)	3.1 ± 3.2	7.9 ± 5.9	<0.001
cN0/N1/N2/N3 (No., %)	58:5:11:0 (78.4:6.8: 14.8:0)	42:9:18:1 (60.0:12.9: 25.7:1.4)	0.042

SD, standard deviation.

**Table 2 cancers-15-04699-t002:** Area under curve and univariable logistic regression analysis of clinical variables for maxillary bone invasion.

Variables	AUC (95%CI)	Odds Ratio (95%CI)	*p-*Value
Age (Continuous)	0.527 (0.432, 0.622)	1.005 (0.983, 1.027)	0.660
Sex (Male to female)	0.589 (0.508, 0.670)	2.067 (1.062, 4.022)	0.033
Tumor location (Upper alveolus to hard palate)	0.547 (0.483, 0.609)	1.896 (0.795, 4.522)	0.149
Pathology	0.566 (0.481, 0.651)		
Squamous cell carcinoma		Reference	
Minor salivary gland carcinoma		2.444 (0.745, 8.017)	0.140
Melanoma		1.730 (0.535, 5.590)	0.360
CT bone invasion (*n* = 144) (Presence vs. absence)	0.779 (0.712, 0.847)	13.049 (5.829, 29.214)	<0.001
PET SUVmax (*n* = 84) (Continuous)	0.714 (0.6015, 0.826)	1.150 (1.036, 1.277)	0.009
Tumor long axis (*n* = 140) (Continuous)	0.748 (0.667, 0.828)	1.842 (1.385, 2.449)	<0.001
Tumor short axis (*n* = 140) (Continuous)	0.745 (0.662, 0.826)	2.147 (1.508, 3.055)	<0.001
Tumor thickness (*n* = 140) (Continuous)	0.723 (0.618, 0.827)	2.360 (1.346, 4.138)	0.001
Tumor area (*n* = 140) (Continuous)	0.753 (0.673, 0.833)	1.137 (1.053, 1.228)	0.003
Lymph node metastasis (cN+ to cN0)	0.592 (0.517, 0.666)	2.417 (1.163, 5.022)]	0.018

AUC, Area under receiver operating characteristics curve; CI, confidence interval.

**Table 3 cancers-15-04699-t003:** MBI prediction models developed by multivariable logistic regression analysis in the patients with HP/UA cancer.

	Predictor Variables	Model 1 (*n* = 144)	Model 2 (*n* = 140)	Model 3 (*n* = 140)	Model 4 (*n* = 84)
Adjusted odds ratio (95%CI)	CT bone invasion	13.049 (5.829, 29.214)	11.616 (4.843, 27.863)	9.631 (3.892, 23.836)	18.146 (4.276, 76.996)
	Tumor long axis (cut-off value = 2.0, or 1.8 cm)		3.153 (1.168, 8.512) ^a^		10.457 (1.345, 81.320) ^b^
	Tumor area (cut-off value = 4.0 cm^2^)			2.997 (1.215, 7.394)	
	Sex (male vs. female)		2.384 (1.003, 5.664)	2.204 (0.932, 5.216]	4.202 (1.092, 16.17)
	Lymph node metastasis (cN+ vs. cN0)		1.806 (0.724, 4.502)	1.738 (0.689, 4.385]	2.505 (0.643, 9.754)
	PET SUVmax (cut-off value = 6.0)				5.519 (1.49, 20.446)
AUC (95%CI)		0.779 (0.712, 0.847]	0.853 (0.790, 0.917)	0.854 (0.790, 0.918)	0.911 (0.847, 0.975)
AUC (95%CI) from internal validation			0.839 (0.687, 0.983)	0.844 (0.695, 0.984)	0.884 (0.758, 0.993)
Brier score ^c^		0.168	0.152	0.152	0.110
*p-*value ^d^		Ref	0.0012	0.0015	0.0002

^a^ Cutoff value of tumor long axis = 2.0 cm. ^b^ Cutoff value of tumor long axis = 1.8 cm. ^c^ Brier score: the value of the Brier score is between 0.0 and 1.0, where a model with perfect prediction has a score of 0.0 and the worst has a score of 1.0. ^d^ Comparison of AUC by DeLong’s test. AUC, area under receiver operating characteristics curve; CI, confidence interval.

**Table 4 cancers-15-04699-t004:** Scoring systems for maxillary bone invasion prediction and assigned points based on regression coefficient.

	Regression Coefficient Ratio	Score
Without PET/CT		
CT bone invasion	4.1	5
Tumor area ≥ 4.0 cm^2^	2.0	2
Male sex	1.4	2
cN+	1.0	1
Total score = 10 (low risk: 0–4, intermediate risk: 5–7, high risk: 8–10)		
With PET/CT		
CT bone invasion	4.2	5
Tumor area ≥ 4.0 cm^2^	1.9	2
Male sex	1.8	2
cN+	1.0	1
PET SUVmax ≥ 6.0	2.2	3
Total score = 13 (low risk: 0–6, intermediate risk: 7–8, high risk: 9–13)		

## Data Availability

The datasets generated and analyzed in the current study are available from the corresponding author upon reasonable request.

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
