# Peer review of "Prediction of Maxillary Bone Invasion in Hard Palate/Upper Alveolus Cancer: A Multi-Center Retrospective Study"

_cancers, 2023, doi:10.3390/cancers15194699_

Round 1

Reviewer 1 Report

The manuscript by Choi N. et al. is interesting and well-done. The authors describe risk factors related to MBI; the methodology is extensive, well explained, and tightly related to the results.
The results are explained in detail and provide a good comparison between the variables studied.
The discussion is well executed and discusses each variable adequately.
I have only one question.
Would it be possible for the authors to review and use the 5th version of the WHO to classify the tumors?

Author Response

Reviewer 1.

Comments and Suggestions for Authors

The manuscript by Choi N. et al. is interesting and well-done.

The authors describe risk factors related to MBI; the methodology is extensive, well explained, and tightly related to the results.
The results are explained in detail and provide a good comparison between the variables studied.
The discussion is well executed and discusses each variable adequately.

I have only one question.
Would it be possible for the authors to review and use the 5th version of the WHO to classify the tumors?

Answer 1.

Thanks for your comment. As your recommendation, we reviewed the 5th version of WHO head and neck tumor classification. We changed the reference from WHO 4th edition to 5th edition. There was no change of pathologic diagnosis between 4th edition versus 5th edition of WHO head and neck tumor pathology classification in subjects of our research.

Correction 1.

Materials and Methods

The pathological diagnosis consisted of squamous cell carcinoma, salivary gland carcinoma, and melanoma. Tumor pathology and histologic grade were classified according to the 5th edition of World Health Organization histological classification [24].

Reference

  1. WHO Classification of Tumours Editorial Board. Head and neck tumours. Lyon (France): International Agency for Research on Cancer; 2022. (WHO classification of tumours series, 5th ed.; vol. 9). https://publications.iarc.fr/

Reviewer 2 Report

This is an interesting study about the prediction of maxillary bone invasion (MBI) in hard palate/upper alveolus (HP/UA) cancer. The authors performed a multi-center retrospective study, enrolling 144 cases. The aim was to design a prediction model for MBI using several radiological and clinical variables of HP/UA cancer.

The topic is original and relevant in the field. Indeed, previous studies focused on mandibular invasion.

The conclusions are consistent with the evidence and arguments presented and address the main question posed. References are appropriate.

The paper is well written. However, some issues remain.

Please better describe how bone invasion at follow-up was identifies.

“Other” tumors should be better specified in table 1.

Author Response

Reviewer 2.

This is an interesting study about the prediction of maxillary bone invasion (MBI) in hard palate/upper alveolus (HP/UA) cancer. The authors performed a multi-center retrospective study, enrolling 144 cases. The aim was to design a prediction model for MBI using several radiological and clinical variables of HP/UA cancer.

The topic is original and relevant in the field. Indeed, previous studies focused on mandibular invasion.

The conclusions are consistent with the evidence and arguments presented and address the main question posed. References are appropriate.

The paper is well written. However, some issues remain.

Please better describe how bone invasion at follow-up was identifies.

Answerer 2-1.

Thanks for the comments. We defined maxillary bone invasion (MBI) in two ways. First, in the patients who underwent bone resection during primary tumor surgery, MBI was defined as pathological cortical/marrow bone invasion. Second, in the patients who did not undergo bone resection during primary tumor surgery, MBI was defined as clinical and radiologic bone invasion during the follow-up period (more than 2 years).

Materials and Methods

Section 2.2.

Clinical bone invasion was estimated by preoperative imaging and physical examination.

Section 2.3.

Radiological CT bone invasion was defined as when a contrast-enhanced primary tumor was identified within the cortical bone and the cortical bone was partially eroded or totally destroyed.

Section 2.5.

Absence of MBI in bone-resected cases was set as true negativity for MBI (n = 25). In addition, for cases without bone resection (n = 73), the absence of MBI was confirmed by the clinical and radiological follow-up more than 2 years (n = 53). Clinical and radiologic bone invasion during follow-up was detected in 24 patients. The final 70 patients were defined as MBI-positive (46 with bone resection and 24 without bone resection).

“Other” tumors should be better specified in table 1.

Answer 2-2.

Thank you for your comment.

Histology was classified into (1) squamous cell carcinoma, (2) minor salivary gland carcinoma, and (3) melanoma. Minor salivary gland carcinoma was further classified into mucoepidermoid carcinoma, adenoid cystic carcinoma, and others.

Others included adenocarcinoma, salivary duct carcinoma, carcinoma ex pleomorphic adenoma, acini cell carcinoma, epithelial-myoepithelial carcinoma. We described the detailed histologic diagnosis in revised Table 1.

Revised Table 1.

Histology (No., %)

Squamous cell carcinoma

27 (36.5)

33 (47.1)

.283

Minor salivary gland carcinoma

37 (50.0)

32 (45.7)

Mucoepidermoid carcinoma

19 (25.7)

12 (17.1)

Adenoid cystic carcinoma

5 (6.8)

14 (20.0)

Carcinoma ex pleomorphic adenoma

4 (5.4)

0 (0)

Adenocarcinoma

3 (4.1)

5 (7.1)

Epithelial-myoepithelial carcinoma

3 (4.1)

0 (0)

Acinic cell carcinoma

2 (2.7)

1 (1.4)

Salivary duct carcinoma

1 (1.4)

0 (0)

Melanoma

10 (13.5)

5 (7.1)

Reviewer 3 Report

Dear authors,

Congratulations on you hard work. I consider this manuscript suitable for publication after minor improvements.

1. Clarification on how patients were selected for follow-up in cases without primary bone resection would be beneficial, as this could influence the determination of true negatives.

2. The authors mentioned that MRI was performed in a subset of cases but did not analyze its diagnostic ability. Including a brief discussion of MRI's role in MBI prediction, even if it's limited due to the subset size, would provide a more comprehensive overview. 

Author Response

Reviewer 3.

Congratulations on your hard work. I consider this manuscript suitable for publication after minor improvements.

  1. Clarification on how patients were selected for follow-up in cases without primary bone resection would be beneficial, as this could influence the determination of true negatives.

Answer 3-1.

Thank you for your comment, and this is an important point.

In general, in cases where bone invasion was evident on clinical findings and preoperative imaging, maxillary bone resection was performed during the primary surgery. However, in cases where bone invasion was absent or ambiguous on preoperative imaging, whether to perform maxillary bone resection is determined by the surgeon taking into consideration factors such as tumor size, staging, reconstruction method and patient morbidity. Therefore, the true negativity of MBI could be underestimated, in cases without primary bone resection.

Therefore, these aspects represented a major limitation of our study, and we mentioned it in the discussion section.

To overcome this limitation (partially), we conducted a secondary analysis of the cases with primary bone resection (Model 5 and 6 in Supplementary Table 2), and confirmed our results in total cases (Model 3 and 4 in Table 3). 

We designed this study to make more accurate clinical decision by surgeons for the management of maxillary bone in HP/UA cancer patients. We believe that this study will be helpful for the future management of maxillary bone in patients with HP/UA cancer.

  1. The authors mentioned that MRI was performed in a subset of cases but did not analyze its diagnostic ability. Including a brief discussion of MRI's role in MBI prediction, even if it's limited due to the subset size, would provide a more comprehensive overview.

Answer 3-2.

Thank you for your comment.

A comparative study of MRI, CT, and PET/CT for the detection of mandibular invasion showed that MRI had the highest sensitivity of the three [15,31]. The authors concluded that any single imaging modalities did not have sufficient diagnostic accuracy, and multiple imaging methods should be considered together for accurate prediction. Likewise, MRI could improve the diagnostic ability of MBI with its higher sensitivity in HP/UA cancer.

However, in this study, MRI was performed only in a subset of our cohort (n = 33), who had an extensive disease, or suspicious bone invasion in CT images. Consequently, those who had preoperative MRI also underwent bone resection and most of them were diagnosed with pathologic bone invasion. This introduced a significant selection bias, which led us to exclude MRI from our analysis.

In short, due to the small number and selection bias of preoperative MRI, we did not analyze the diagnostic ability of MRI for the HP/UA bone invasion. This point should be re-evaluated in future studies.

[END]